



# Iceberg topography and volume classification using TanDEM-X interferometry

Dyre O. Dammann[1,2], Leif E.B. Eriksson[1], Son V. Nghiem[3], Erin Pettit[4], Nathan T. Kurtz[5], John G. Sonntag[5], Thomas Busche [6], Franz J. Meyer[7], Andrew R. Mahoney[7]

[1]Department of Space, Earth, and Environment, Chalmers University of Technology, Gothenburg, Sweden
[2]StormGeo, Bergen, Norway
[3]Jet Propulsion Laboratory, California Institute of Technology, Pasadena, CA, USA
[4]College of Earth, Ocean, and Atmospheric Sciences, Oregon State University, Corvallis, OR, USA
[5]National Aeronautics and Space Administration Goddard Space Flight Center, Greenbelt, MD, USA
[6]German Aerospace Center, Microwaves and Radar Institute, Oberpfaffenhofen, Germany
[7]Geophysical Institute, University of Alaska Fairbanks, Fairbanks, AK, USA

*Correspondence to*: Dyre O. Dammann (dyre.dammann@stormgeo.com)

**Abstract.** Icebergs in polar regions affect water salinity, alter marine habitats, and impose serious hazards on maritime operations and navigation. These impacts mainly depend on the iceberg volume, which remain an elusive parameter to measure. We investigate the capability of TanDEM-X bistatic single-pass synthetic aperture radar interferometry (InSAR) to derive iceberg subaerial morphology and infer total volume. We cross-verify InSAR results with Operation IceBridge (OIB) data acquired near Wordie Bay, Antarctica, as part of the OIB/TanDEM-X Antarctic Science Campaign (OTASC). While icebergs are typically classified according to size based on length or maximum height, we develop a new volumetric classification approach for applications where iceberg volume is relevant. For icebergs with heights exceeding 5 m, we find iceberg volumes derived from TanDEM-X and OIB data match within 7 %. These results suggest that TanDEM-X could pave the way for future single-pass interferometric systems for scientific and operational iceberg mapping and classification.

## 1 Introduction

Icebergs play an important role in polar oceans by providing habitat for marine mammals (Blundell et al., 2011;Lydersen et al., 2014), enhancing local primary production (Smith et al., 2007), facilitating sea ice growth with fresher meltwater (Merino et al., 2016), hindering advection of sea ice (Massom et al., 2001;Arrigo and van Dijken, 2003), and modifying water properties in the upper layer of the ocean within fjords (Moon et al., 2018) as well as in the open ocean (Gladstone et al., 2001;Silva et al., 2006). Icebergs also impact local weather conditions and their influence on ocean conditions are linked to carbon cycling and climate (Helly et al., 2011). Furthermore, polar oceans are opening up to more maritime activities while icebergs are expected to become numerous and thereby exacerbate risks to shipping and offshore activities (Eik and Gudmestad, 2010;Bigg et al., 2018). Icebergs are found in high concentrations near the outlets of marine-terminating glaciers and ice sheets (i.e., tidewater glaciers and ice shelves), especially in marginal seas of the Antarctic and Greenland Ice Sheets, and also in significant quantities in Alaska and Chile. Icebergs are created by calving events at the glacier ice-ocean boundary and can range in size



from less than a meter length at the waterline to over 100 km e.g. (Lazzara et al., 1999;Parmiggiani et al., 2018). Icebergs differ largely in shape based on age and geographic region. For instance, the largest icebergs (typically those greater than a few kilometers) break off as tabular icebergs from floating ice shelves.

While icebergs are more common near ice-ocean boundaries, they drift across long distances with ocean currents and thus
pose potential hazards even hundreds of kilometers away from any glacier (Schodlok et al., 2006). One such location is the western part of the North Atlantic Ocean where icebergs are regularly transported south through the Davis Strait into high-traffic North Atlantic shipping lanes (Kollmeyer, 1978). Along this path, icebergs have been extensively surveyed (Jacka and Giles, 2007;Romanov et al., 2012) and are continuously tracked by the International Ice Patrol (IPP) (Murphy and Cass, 2012) according to size (Table 1) and shape (e.g. tabular vs. non-tabular). Despite the importance of these reports, they are limited
in both temporal and spatial extent, and are largely unavailable in polar regions. Remote sensing techniques such as LiDAR (Scambos et al., 2005) and optical stereo photogrammetry (Enderlin and Hamilton, 2014) have been used to evaluate both the vertical and horizontal extent of icebergs as tools with larger spatial coverage. However, microwave remote sensing systems e.g. radar altimeter, (Tournadre et al., 2012) and scatterometer, (Stuart and Long, 2011) are advantageous due to independence from light and weather conditions.

Synthetic aperture radar (SAR) has shown to be a robust tool to detect smaller icebergs with a resolution down to the meter scale (Willis et al., 1996;Williams et al., 1999;Silva and Bigg, 2005). SAR-based methods are typically based on intensity thresholds based on the assumption that icebergs return a stronger backscatter signal than their surroundings (Willis et al., 1996;Silva and Bigg, 2005;Wesche and Dierking, 2012). More recently, object-based image analysis with a focus on classifying objects rather than individual pixels has also shown promise (Mazur et al., 2017). Techniques based on backscatter
intensity alone, however, are suboptimal where the brightness between icebergs and background is similar; for example, when icebergs are surrounded by wind-roughened water (Willis et al., 1996;Wesche and Dierking, 2012). Such techniques can potentially be improved by polarimetric SAR (Dierking and Wesche, 2014). However, standard SAR-based approaches provide information strictly pertaining to the nadir-view two-dimensional shape and concentration of icebergs, but do not provide information pertaining to the height necessary to fully classify the icebergs according to Table 1.

The evaluation of height is not only necessary for classification, it is also crucial for assessing the three-dimensional surface morphology relevant to marine mammals (McNabb et al., 2016) and the stability of the iceberg (Guttenberg et al., 2011). Moreover, iceberg morphology is related to the overall iceberg volume relevant to a number of important properties such as drift and decay (Hamley and Budd, 1986;Barker et al., 2004;Jacka and Giles, 2007;Crawford et al., 2018), freshwater contribution (Jacobs et al., 1992;Silva et al., 2006;Enderlin and Hamilton, 2014;Moon et al., 2018), and icebergs as a hazard
(Fuglem and Muggeridge, 1999). The mass and thus mechanical properties (Romanov et al., 2012) and potential impact on structures and vessels (Liu et al., 2011) are also directly related to iceberg volume. The total volume of icebergs is difficult to

estimate unless the subaerial morphology is measured. The physical limitations of existing techniques in measuring morphology hamper consistent and accurate iceberg evaluation over large areas (Romanov et al., 2017).

We investigate TanDEM-X SAR interferometry (InSAR) (Rosen et al., 2000) as an alternative approach to assess iceberg topography and for estimating iceberg volume. This technique to extract topography from the phase information from two complex SAR scenes has previously been utilized to assess sea ice ridge topography (Dammann et al., 2017;Dierking et al., 2017). However, InSAR has only briefly been explored to acquire information on iceberg topography (Power et al., 2011;Zakharov et al., 2013;Zakharov et al., 2017) and topographic changes (García et al., 2012). We therefore explore accuracy, applications, and limitations of InSAR-derived surface morphology estimates of icebergs. We also develop a size classification scheme based on iceberg volume, which is more directly relevant to certain end users than present approaches.

## 2 Data and methods

### 2.1 TanDEM-X data and study area

This work utilizes data from the twin constellation TanDEM-X, an X-band SAR system operating since 2010. Each individual satellite of the TanDEM-X constellation has a repeat-pass cycle of 11 days and an orbit design optimized for constellation flight allows simultaneous acquisitions between the two constellation partners. TanDEM-X is operated by the German Aerospace Center (DLR) and is currently the only system which can acquire SAR imagery with temporal lag on the order of milliseconds necessary to reliably and consistently evaluate topography of non-stationary surfaces undergoing slow motions. The system features two X-band ($\lambda = 3.1$ cm) SAR sensors resulting in m-scale resolution imagery.

This work utilizes a TanDEM-X bistatic acquisition from 29 Nov. 2017 at 00:32:09.874 UTC over Buffer Island in Wordie Bay on the west side of the Antarctic Peninsula (Figure 1). This is the region of the former Wordie Ice Shelf, which broke away from shore between 2008 and 2009. The region is often populated by numerous icebergs, ranging from a few meters to several hundred meters in width, frozen into the landfast sea ice as seen from an Operation IceBridge flight on 21 November 2017 (Figure 2). The acquisition was taken in ascending orbit #58160 with an incident angle of 38.5˚. The acquisition features two dual polarization (HH and VV) stripmap images with a swath width of 15 km and a time lag of 10 ms. The perpendicular and along-track baseline between images was 154 m and 151 m respectively. In the case used here, the icebergs were confirmed stationary, frozen into the landfast ice, by consecutive TanDEM-X overpasses 11 days apart.

### 2.2 Interferometric SAR processing

InSAR is a technique utilizing two complex SAR scenes where the resulting interferogram represents the phase difference, $\Delta\Phi$, between the two scenes and is represented by phase values in the range [-π, π[. For TanDEM-X data with large spatial and short temporal baselines, the observed values for $\Delta\Phi$ is largely attributed to phase component due to topography $\Delta\Phi_{topo}$:



$$\Delta\Phi_{topo} = \frac{4\pi B_{\perp} h}{\lambda R \sin\theta} \qquad (1)$$

where $R$ is the slant range, $B_{\perp}$ is the perpendicular baseline, $\theta$ is the incident angle, $\lambda$ is the wavelength, and $h$ is the topographic height. If the height exceeds a certain threshold called the height of ambiguity, the phase will wrap around from -$\pi$ to $\pi$ causing phase ambiguities. The height of ambiguity can be expressed:

$$h_a = \frac{\lambda R \sin\theta}{2 B_{\perp}} \qquad (2)$$

For the image used here, the height of ambiguity is $h_a = 41.8$ m.

We processed the VV channel of the complex TanDEM-X scene for backscatter intensity and phase-derived height using the Gamma Software (Werner et al., 2000). For backscatter, this involved multilooking of 2x2 pixels (resulting in resolution of roughly 2.7 m in range and 4.7 m in azimuth) and filtering 5x5 pixels using a standard boxcar filter. To obtain phase-derived height (referred to hereafter as InSAR Digital Elevation Model or InSAR DEM), we followed a standard InSAR processing workflow (Bamler and Hartl, 1998). This processes involves multilooking (2x2 pixels) and adaptive phase filtering (Goldstein and Werner, 1998) to reduce phase noise. Here, we used a relatively small filtering FFT window size of 8. The small window was chosen to preserve as much detail of the icebergs as possible. Finally, we geocoded the backscatter image and interferogram in a UTM 17S projection (with a WGS84 datum in an ellipsoidal reference height system) with 2.5m square pixel spacing.

The theoretical relative height accuracy of the InSAR DEM can be calculated as:

$$\sigma_h = \frac{\lambda}{4\pi} \frac{R \sin\theta}{B_{\perp}} \sigma_{\phi} \qquad (3)$$

where $\sigma_{\phi}$ is the standard deviation of the InSAR phase estimate, which is expressed:

$$\sigma_{\phi}^2 \approx \frac{1}{2N_L} \frac{1-\gamma^2}{\gamma^2} \qquad (4)$$

in which $N_L$ is the independent number of looks and $\gamma$ is the interferometric coherence (Rosen et al., 2000; Dierking et al., 2017). Based on an average coherence, $\gamma \sim 0.7$ for our study area, this results in a relative height accuracy, $\sigma_h \sim 2.5$ m.

## 2.3 Iceberg classification

We discriminated icebergs from surrounding ice by applying the InSAR DEM. Our method requires a minimum iceberg height of 5 m, twice that of the height accuracy $\sigma_h$, which means we cannot detect growlers and bergy bits (Table 1). Setting such a high threshold results in minimized false positives from phase noise. Once an initial iceberg mask was created via thresholding,



we performed a subsequent geometric opening operation to remove noise in the initial mask by removing objects of less than roughly 10x10 pixels in size.

We classified the delineated icebergs in multiple ways. First, we use the classification outlined in Table 1: small, medium, large, and extra-large according to equivalent length (square root of total area) and height. Second, we classified them as tabular or non-tabular through the ratio of height above water to equivalent length because tabular icebergs have a smaller height compared to their length. Tabular icebergs are those that calve from floating ice shelves or tongues that are rectangular cuboid shape and of stable geometry such that they do not flip from their original orientation. Because of the nature of their formation, tabular icebergs are typically "medium" height (Table 1), but "extra-large" length, as the stability for their shape requires a minimum length of approximately 5 times the height (Bass, 1980) . A tabular iceberg classification is important as it both impacts the predicted keel depth and provides information on the iceberg source location (tabular icebergs are created from floating termini and ice shelves). Tabular icebergs eventually decay through melt and fracture into shapes belonging to the non-tabular category.

Third, we calculate the iceberg volume. The iceberg subaerial (above sea level) volume can be calculated by integrating the phase-derived height above a reference surface within the delineated areas occupied by icebergs. Integration steps equal the pixel spacing of 2.5 m. The total iceberg volume then can be inferred from this subaerial volume, which is 11% of the total volume (assuming an ice density of 917 $kg/m^3$ for ice and 1030 $kg/m^3$ for sea water). And finally, we estimated the minimum, expected, and maximum keel depths (draft) based on iceberg stability analyses e.g., Bass (1980), limited measurements (Barker et al., 2004), and similar analyses (Sulak et al., 2017). We defined a minimum keel depth as that for tabular icebergs d=V/A, assuming a rectangular cuboid. We defined extreme maximum keel depth as an inverted pyramid or cone d=3V/A; however, this shape is unlikely to persist due to rapid melting of a pointed keel. We therefore determine a more realistic "expected" keel depth based on the data presented by Barker et al. (2004) suggesting d=2.91L$^{0.71}$.

## 2.4 Validation data

The TanDEM-X data over Wordie Bay was acquired in conjunction with the Operation IceBridge (OIB) fall 2017 Antarctic campaign on November 21, 2017, of which the OIB/TanDEM-X Antarctic Science Campaign (OTASC) (Nghiem et al., 2018) was a component. OIB is a decade-long series of annual Arctic and Antarctic airborne surveys intended to bridge the gap between polar land and sea ice measurements collected by NASA's ICESat-1 and ICESat-2 spacecraft (Zwally et al., 2002;Markus et al., 2017). The aircraft used for the 2017 Antarctic campaign was NASA's P-3 Orion, a long-range, four-engine turboprop capable of flights of up to ten hours in length, at low altitude, and at speeds of 250 knots. For this campaign the aircraft was based in distant Ushuaia, Argentina, owing to the lack of suitable air basing facilities for such a large aircraft on the Antarctic continent. This arrangement limited the on-site survey time available.



For the OTASC-coordinated flights, OIB selected suitable TanDEM-X ground tracks for the day of each flight and coordinated the aircraft's arrival on the track to be as close in time to that of the spacecraft as practical. The aircraft was equipped with a suite of geophysical instruments, including a pair of scanning laser altimeters, known as the Airborne Topographic Mapper, or ATM (Martin et al., 2012), a digital high-resolution camera system called the Digital Mapping System (DMS) (Dominguez, 2010), and associated GPS, inertial, and precise navigation systems. When merged, the three-dimensional point cloud data from the ATM and the geolocated imagery from the DMS enabled the construction of a 250 m wide, sub-meter resolution digital elevation model (DEM) with a vertical accuracy of ~0.2 m and 10x10 cm pixel spacing (Nghiem et al., 2018). Data were acquired over our study area in Wordie Bay on 21 November, 2017 (17:26:58 - 17:28:34 UTC) with acquisition ID 172759 (Studinger, 2014), eight days prior to the TanDEM-X acquisition. The data were re-projected from WGS84 Polar Stereographic to WGS84 UTM 17S, equal to that of the geocoded SAR data with a pixel spacing of 0.5 m. The model is referred to from here as DMS DEM.

## 3 Results

We utilized a roughly 15x15 km section of the TanDEM-X acquisition centered around Buffer Island. We processed the scene for backscatter intensity (Figure 3a) and phase-derived height (InSAR DEM) (Figure 3b). These data exhibit large amounts of stationary icebergs of different sizes enclosed by landfast sea ice. We compared the TanDEM-X backscatter and InSAR DEM with the DMS DEM. This comparison was done within a small, roughly 10 km$^2$ subset (red rectangle in Figure 3) around the validation data (Figure 4a and b). The DMS DEM dataset follows a curved path covering many small and parts of some larger icebergs (Figure 4c). For validation, we defined four transects (T1 to T4) and six areas (A1 to A6) completely situated within the boundaries of the DMS DEM. These are spread throughout the dataset and cover different icebergs and iceberg sizes.

Transects T1, T2, and T4 cross larger icebergs with heights exceeding 20 m, while T3 crosses smaller icebergs with heights below 10 m. For all transects, the DMS DEM compares reasonably well with the InSAR DEM as they follow generally the meter-scale topography (Figure 5). However, there are several outliers along the transects resulting in an average root mean square error (RMSE) of 3.46 m (ranging between 4.64 m for T1 and 2.23 m for T3). First, the DMS DEM is smoother than the InSAR DEM, most likely due to (1) phase noise equivalent to $\sigma_h \sim 2.5$ m and (2) substantial averaging of the DMS DEM product from its original pixel spacing. Second, there are discrepancies between the InSAR DEM and DMS DEM at the location of rapid elevation changes with vertical or steep slopes (see circled areas in Figure 5). This discrepancy is likely due to layover (compression of targets closest to the satellite) effects, increasing elevation in the InSAR DEM on the side of the iceberg facing the satellite and vice versa. Third, two areas feature substantial variability of over 10 m in the InSAR DEM not represented in the DMS DEM (see purple arrows in Figure 5). These areas correspond to low backscatter (see black lines in Figure 5) likely as a result of radar shadowing. The low backscatter can significantly reduce coherence through low signal-to-noise ratios and thus height accuracy according to Equation 4.



The backscatter profiles shown in Figure 5 exhibit variability somewhat correlated with the location of icebergs (i.e. areas of sea ice in between icebergs corresponds to reduced backscatter and icebergs correspond to relatively high backscatter), which has been observed before (Willis et al., 1996;Silva and Bigg, 2005;Wesche and Dierking, 2012). However, there is not a direct relationship between backscatter and iceberg elevation in our data set. We quantified the low correlation between the two in

our data by estimating correlation coefficient between backscatter and DMS DEM for areas A1 to A6 (not shown). These low correlations (mean R=0.58) are expected because surface roughness and slope dominate radar brightness, rather than the elevation of icebergs. Backscatter, therefore, is not sufficient to infer either iceberg height or volumetric size.

We similarly compared the DMS DEM and InSAR DEM for areas A1 to A6 (Figure 6) and show higher correlations between the DMS DEM and InSAR DEM (mean R=0.69) than the DMS DEM and backscatter.  The datasets show close to a 1:1 linear

trend with R values exceeding 0.6. However, there are substantial outliers for the reasons pointed out previously, leading to RMSE values ranging between 1.89 m (A1) and 7.08 m (A5). To further the understanding of these outliers, we calculated the differences between the DMS DEM and the InSAR DEM. We examined the most significant differences near A3 to A6 (see Figure 7). These differences indicate that substantial offsets are located around steep vertical sides of icebergs (see circled area in Figure 7 and circles in Figure 5). This cannot be attributed to phase unwrapping errors as offsets typically do not exceed $h_a$.

Neither are they likely to be attributed to an offset between DEMs as such offsets are not systematically occurring in similar locations. As the offset occurs in a region of low backscatter values, reduced coherence is the most likely source of the observed differences. Despite the inaccuracies in areas of low backscatter, the InSAR DEM compares well with the DMS DEM, indicating that the InSAR DEM sufficiently captures the surface morphology of icebergs and enables the derivation of iceberg volume.

We calculated the total volume of ice above a reference surface (i.e. zero elevation calibrated to the lowest InSAR DEM values i.e. left corner of Figure 3b) for A1 to A6. The total volumes calculated from the DMS DEM and InSAR DEM for each area are listed in Table 2. The iceberg sizes enclosed by the boundary of areas A1 to A6 vary greatly from having m-scale relief to several tens of meters, and thus greatly vary in total iceberg volume and relative volume accuracy. For instance, A1 contains the lowest volume of icebergs, but features the largest discrepancy (measured in %) between the DEMs. The largest iceberg

in terms of height is found in A5, which has the best match between DEMs. These results indicate that volume is not captured well for small icebergs such as growlers and bergy bits and volume estimates are in general most accurate for larger icebergs. However, comparing A2 and A3 reveals exceptions from a direct relationship between height and accuracy.



## 4 Discussion

### 4.1 Volume classification

We demonstrate here the advantages of using TanDEM-X for evaluating icebergs using InSAR. We show that TanDEM-X data enables assessment of subaerial morphology and volume of icebergs with m-scale resolution in a cost-effective manner
in comparison with air reconnaissance. We found that InSAR-derived volumes agree with estimated volumes based on the OTASC data within 7% except for icebergs with a small topographic relief ranging from a few centimeters to meters (barely visible in the interferometric phase) where the volumetric difference was found to be 23%. InSAR-based iceberg assessments can thus potentially be used to enhance understanding of the evolution of icebergs and their impact on local ecosystems. SAR signals are not relying on daylight or weather conditions. Hence InSAR-derived measurements could also potentially be used
in an operational setting for tactical decision-making.

We furthermore explored the potential for classifying icebergs according to volume and how such classification differs from standard approaches. We initially classified icebergs using the InSAR DEM according to the International Ice Patrol (IIP) thresholds of length and height (Table 1). This results in classification of small, medium, large, and extra-large icebergs (Figure 8a). However, with this classification method, only medium, large, and extra-large icebergs are present in our study area. We
further classified icebergs based on height criteria (Figure 8b). This step results in strictly small and medium icebergs being present. Here, icebergs that we classify as extra-large in Figure 8a are medium according to Figure 8b. One likely explanation for this difference is that iceberg shape varies greatly between the region surveyed by IIP near the Grand Banks and our study region in Antarctica dominated by tabular icebergs. This suggests that a one-dimensional metric is suboptimal to describe icebergs.

We classified icebergs according to their derived total volume (Figure 8c). The classification thresholds were chosen with close to equal volumetric bin sizes of about $50 \times 10^6$ m$^3$ with the exception of the smallest icebergs, enabling an incorporation of all icebergs in our study region into four classes. In general, larger icebergs in terms of surface area (Figure 8a) are also classified as larger in terms of volume (Figure 8c) and vice versa. This is expected since iceberg height is limited by its horizontal extent to remain stable and not flip over on its side. Tabular icebergs have larger volumes relative to their height
than their non-tabular counterparts. Figure 8d shows the distribution of tabular icebergs in our study area, identified according to a length to height ratio of 5. The value of our proposed volumetric classification scheme is its application in areas where iceberg volume or mass is of direct relevance to important properties and processes. In a final alternative approach, we classified icebergs according to minimum, expected, and maximum keel depth (Figure 9). This type of classification has relevance for iceberg interactions with the sea floor such as grounding and impacts on subsea installations.





## 4.2 Method constraints and limitations

Even with its potential advantages, TanDEM-X has limitations for the task of iceberg detection and classification due to low data availability in particular over ice-covered waters. The 11-day repeat-cycle is also a disadvantage as it reduces the potential of TanDEM-X for monitoring of icebergs with significant drift speeds. Even so, volume evaluation using TanDEM-X has a

synergistic potential in complimenting existing products. For instance, a single InSAR pair from this mission could be used to identify icebergs and estimate their size and volume. The icebergs could then be tracked through time using other SAR systems, such as Sentinel-1, improving temporal coverage. The accuracy of TanDEM-X in deriving iceberg height critically depends on the perpendicular baseline (Equation 3), which may be suboptimal depending on location and time. The primary goal of the TanDEM-X mission and ongoing operations is to acquire a DEM over land. Therefore, baselines and resulting height of

ambiguities may not be optimal over polar oceans for evaluation of icebergs. This was the case for the OTASC campaign where height of ambiguities ranged between 40-50 m (~150 m perpendicular baseline) making it difficult to evaluate growlers with a subaerial vertical extent of less than 1 m. A height of ambiguity of less than 10 m would have been preferred for this application, but was only possible during the TanDEM-X Science Phase in 2015 (Dammann et al., 2017;Dierking et al., 2017).

Beyond the constraints related to TanDEM-X data, there can be inherent environmental limitations impacting the

interferometric processing and analysis. Examples are situations where icebergs are in a state of significant drift or situations where significant penetration of the SAR signal into the freshwater ice occurs. Atmospheric effects, coregistration errors, and phase changes due to surface change or deformation can also theoretically result in a phase uncertainty, but are unlikely to be significant for bistatic acquisitions. The interferometric phase, $\Delta\Phi$, is not only sensitive to topography, but also surface motion resulting in $\Delta\Phi_{disp}$. Displacement in line-of-sight direction ($\Delta r_{LOS}$) results in a phase change according to $\Delta\Phi_{disp} =$

$4\pi \Delta r_{LOS}/\lambda$ (Dammann et al., 2016). Ice drift can potentially reach close to 1 m s$^{-1}$, which for bistatic mode with 10-ms temporal baseline can result in a phase change $\Delta\Phi_{disp}\sim$ 4 radians. With a height of ambiguity of tens of meters, the phase contribution from motion is significant, but can potentially be removed. For instance, if the iceberg is surrounded by drifting sea ice, the icebergs may possibly drift at comparable speeds if frozen within the sea ice. If not frozen in, the icebergs may drift with different speeds than surrounding sea ice, including the possibility of drift in in the opposite direction if deeper

currents drive iceberg drift. In such a case, the phase of the surrounding sea ice can be used to calibrate roughly zero elevation independent of speed. In the absence of sea ice or in the case of non-homogenous drift, the outer perimeter of the iceberg can also sometimes be used for calibration. This is however difficult if the sides of the iceberg are steep.

When comparing phase-derived height from TanDEM-X with the validation dataset, it is necessary to consider possible significant horizontal or vertical mismatch between the datasets. In this work we strictly geocoded the TanDEM-X data based

on orbit position. Post geocoding, we slightly shifted the DMS DEM (translation) to visually match the InSAR DEM and calibrated the height of both datasets to zero height in an area of no icebergs. We did not perform an absolute height calibration, which inevitably results in remaining inaccuracies such as spatial translational and rotational offsets. Such small positioning





offsets of the InSAR DEM might cause large (> 10 m) height offsets near the edges of icebergs possibly contributing to the offsets seen in Figure 7. Spatial offsets can be reduced by the use of external control points from IceSat or tie points of neighboring TanDEM-X tracks to improve the InSAR DEM referencing.

It is necessary to consider possible penetration of the X-band SAR signal. Both laser altimetry (ATM) and optical photogrammetry (DMS) generally result in a DEM of the ice or snow surface. Ice topography as derived using InSAR on the other hand, is not necessarily the topography of the ice surface, but rather reflects the elevation of the interferometric phase center (Rignot et al., 2001). SAR signals may significantly penetrate into the ice and reflect off subsurface layers or impurities (e.g. air bubbles, fractures) within an iceberg. Little is known about the exact penetration depth of X-band SAR, but it decreases with rising temperature and water content (Gardelle et al., 2012;Hall, 2012). TanDEM-X was shown to penetrate up to 7 m into firn and glacier ice (Millan et al., 2015). In Antarctica, Davis and Poznyak (1993) measured penetration depths at 10 GHz to reach between 2.1 and 4.7 m, and Surdyk (2002) reported a 4 m penetration depth at 10.7 GHz into ice at –88˚C (Gardelle et al., 2012). The penetration of X-band SAR into icebergs may be substantially lower as icebergs are subjected to saline water, warmer surface and internal temperatures close to that of the surrounding water, and contain limited fern, which significantly impacts penetration in glacier studies. For exposed ice during warmer parts of the year, Rignot et al. (2001) reported no significant penetration (±1-2 m) for exposed ice near Jakobshavn Glacier, Greenland. Prior to the acquisition used here, the temperature at the San Martín Base roughly 110 km to NNE of Buffer Island (Figure 1) was roughly +2˚C and remained above freezing for prior 12 hours. The resulting X-band penetration depth based on the warm ice/snow surfaces is therefore likely insignificant (Figure 5).

The classification methods described here are associated with limitations. It has proven difficult in this work to classify bergy bits and growlers based on their modest subaerial relief. For the small to extra-large icebergs that can be classified, the derived volume is based on assumptions that the icebergs are in hydrostatic equilibrium. Close to the coast, as is the case here, it will be necessary to assess bathymetric data to assess the validity of that assumption based on phase-derived height, an approximate sail-to-keel ratio, and bathymetric information. We estimated approximate keel depths based on derived iceberg volume and assumed hydrostatic equilibrium (Figure 9). For the extra-large icebergs in this work, the difference between the estimated minimum and maximum keel depths can reach upwards of 100 m. This can make it problematic to determine whether icebergs are floating or grounded in locations where bathymetric depth falls within this window. Also, keel depth estimates are based on the volume or equivalent length of the iceberg; if two icebergs are connected underwater, then keel depths can be larger than would be calculated by treating the two subaerial parts as individual icebergs. While such situation may not critically impact estimates such as potential freshwater contribution of icebergs, they could lead to significant errors of estimated keel depth, iceberg drift and decay, and their hazard potential for maritime installation.



## 5 Conclusion

In this work, we applied bistatic InSAR data from the TanDEM-X mission over Wordie Bay, Antarctica, to derive topography and volume information of icebergs. We initially validated the phase-derived height of icebergs with elevation data acquired from the OIB/TanDEM-X Antarctic Science Campaign (OTASC). This validation demonstrates that bistatic interferometry

can be a valuable tool in assessing iceberg morphology and volume. We furthermore classified icebergs based on volume. Iceberg volume incorporates both the height dimension and the length scale and is likely an advantageous metric for a number of applications. For instance, volume is of key relevance not only to offshore operations and ice management, but also in the context of marine habitat and ecosystem mapping and glaciology research. Based on derived volume, we were able to estimate the tabular nature of icebergs and bracket minimum and maximum keel depths, resulting in a range of bathymetric depths

where the surveyed icebergs can potentially interact with the sea floor. A detailed discussion of the main uncertainties affecting these estimates was provided. Further work is needed to investigate how to most effectively and accurately classify icebergs using InSAR for different applications.

TanDEM-X is the only current spaceborne SAR system that can be used to consistently evaluate the morphology and volume of icebergs. TanDEM-X has relatively narrow achievable swath width in Stripmap mode (~30 km for single polarization and

~15 km for dual polarization) which is suboptimal for iceberg monitoring across large spatial scales. Future systems with High Resolution Side Swatch Mode may be able to overcome this limitation. TanDEM-X has a repeat-pass cycle of 11 days, limiting data availability and reduces the applicability of InSAR in an operational setting. Future systems such as the LOTUSat-1 (2021 launch) and LOTUSat-2 (2025) X-band SAR missions (Pham, 2017) may alleviate this concern. However, the capability of LOTUSat for iceberg detection needs to be evaluated specifically based on LOTUSat SAR characteristics and orbit

configurations. The planned TanDEM-L mission (Moreira et al., 2015) may further contribute synergistic data in time and space; however, it is unsure how L-band SAR would differ from X-band in assessing iceberg volume due to larger penetration depth. Although the data availability of TanDEM-X is suboptimal for effective ice management and support of operations, volume products could still be a great asset in conjunction with other SAR systems. We also argue that TanDEM-X has the potential to be a valuable tool for deriving the morphological statistics of icebergs to complement volume estimates by other

observation methods. Such statistics could provide value for calving estimates, iceberg drift modeling, and habitat mapping.

## Acknowledgments

This work was supported by the Swedish National Space Agency (Dnr 192/15). TanDEM-X data were provided free of charge by the German Aerospace Center (DLR) through science proposals (XTI_GLAC6921 and XTI_GLAC7297). Operation IceBridge data were provided by the National Aeronautics and Space Administration (NASA).  The research carried out at the

Jet Propulsion Laboratory (S. V. Nghiem), California Institute of Technology, was supported by the NASA Cryosphere Science Program and in part by the NASA Earth Science R&A Program.




## Author Contributions

Dyre Dammann conducted the interferometric processing and analysis and drafted the initial manuscript. Leif Erikson and Son Nghiem provided critical guidance on all aspects of the analysis and manuscript. Son Nghiem, Nathan Kurtz, John Sonntag, and Thomas Busche contributed to the thorough planning and successful execution of the OIB/TanDEM-X Antarctic Science

Campaign, providing satellite and aircraft data used in this research. Erin Pettit provided valuable expertise relevant to icebergs and iceberg classification. Franz J Meyer and Andrew Mahoney contributed guidance in terms of interferometric analysis and interpretation. All co-authors also provided valuable recommendations and corrections resulting in the final manuscript.

## Competing interests

The authors declare that they have no conflict of interest.

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



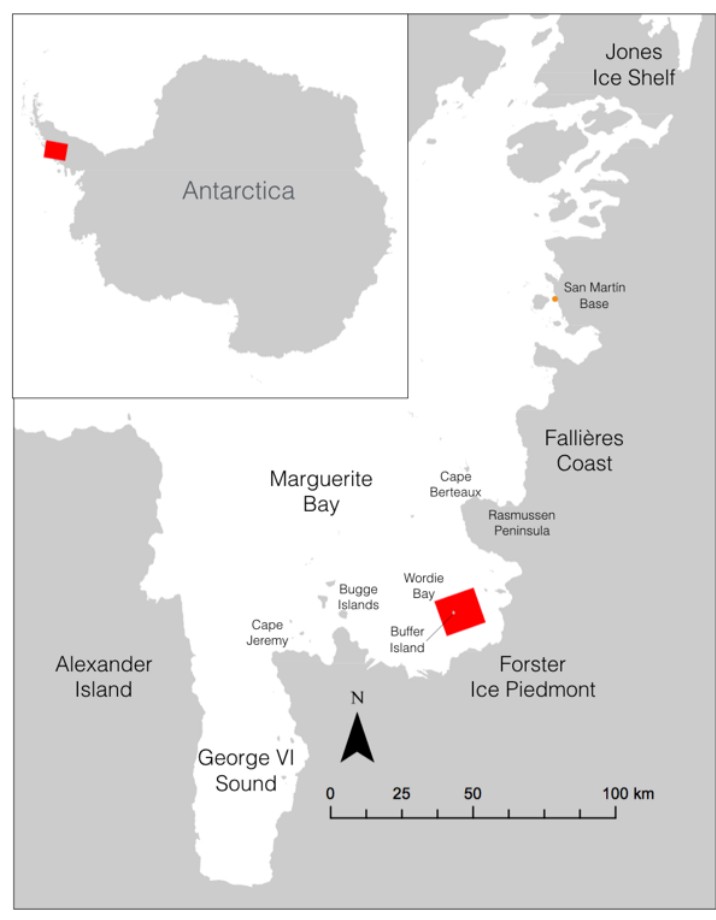

**Figure 1: Study area situated around Buffer Island in Wordie Bay, Antarctica**





**Figure 2: Photograph of Wordie Bay on 21 November 2017 taken at 69.230 S and 68.430 W. The old broken-up Wordie Ice Shelf remains are in the foreground. Photo credit John Sonntag.**



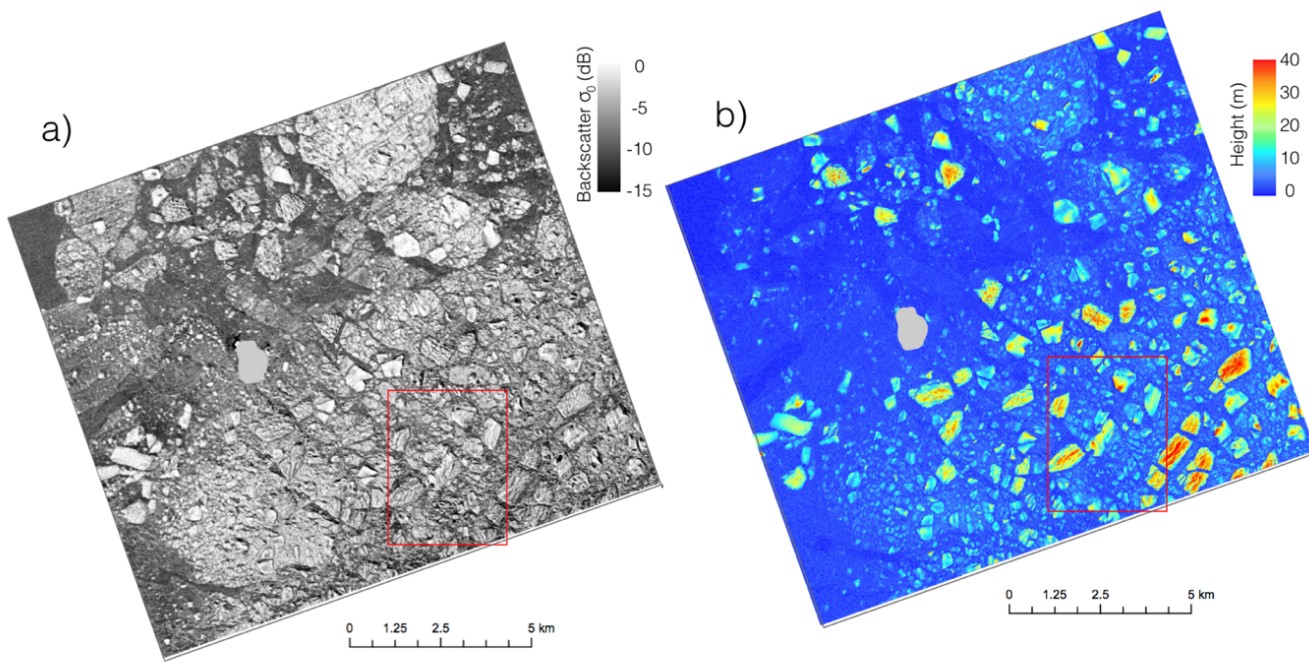

**Figure 3: Cropped area of the TanDEM-X scene processed for backscatter intensity (a) and interferometric height (InSAR DEM) (b). Red rectangle signifies validation area. Buffer Island is masked out in light gray.**



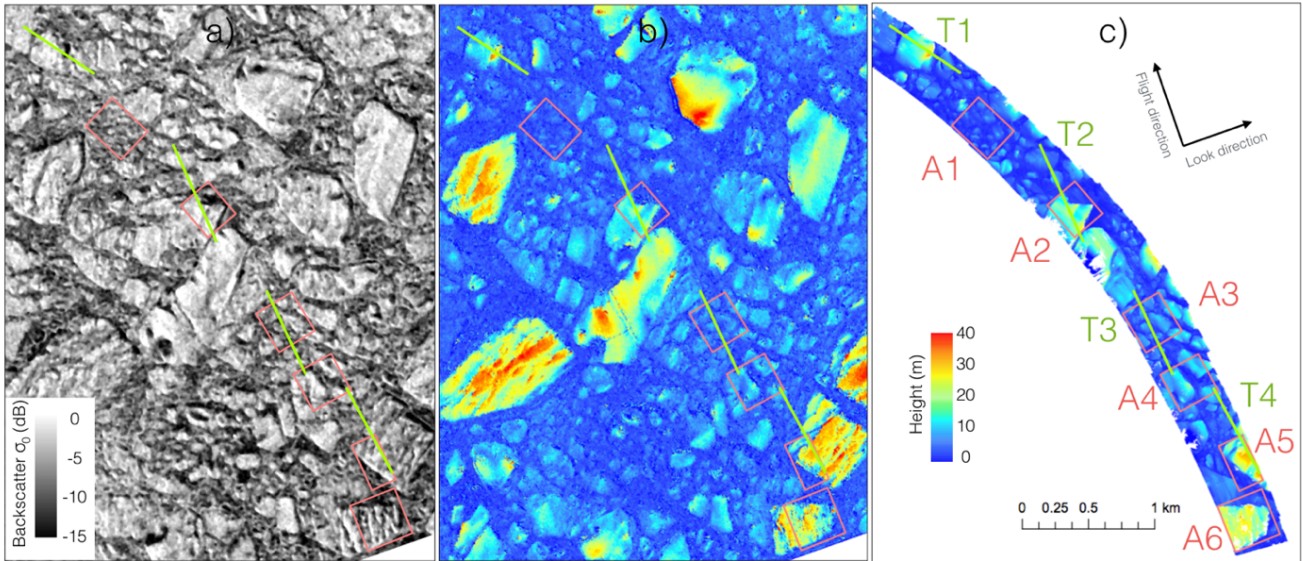

**Figure 4: Backscatter (a) and InSAR DEM (b) over validation area. (c) Validation DEM (DMS DEM). Red rectangles signify individual validation areas. Green lines signify validation cross sections.**



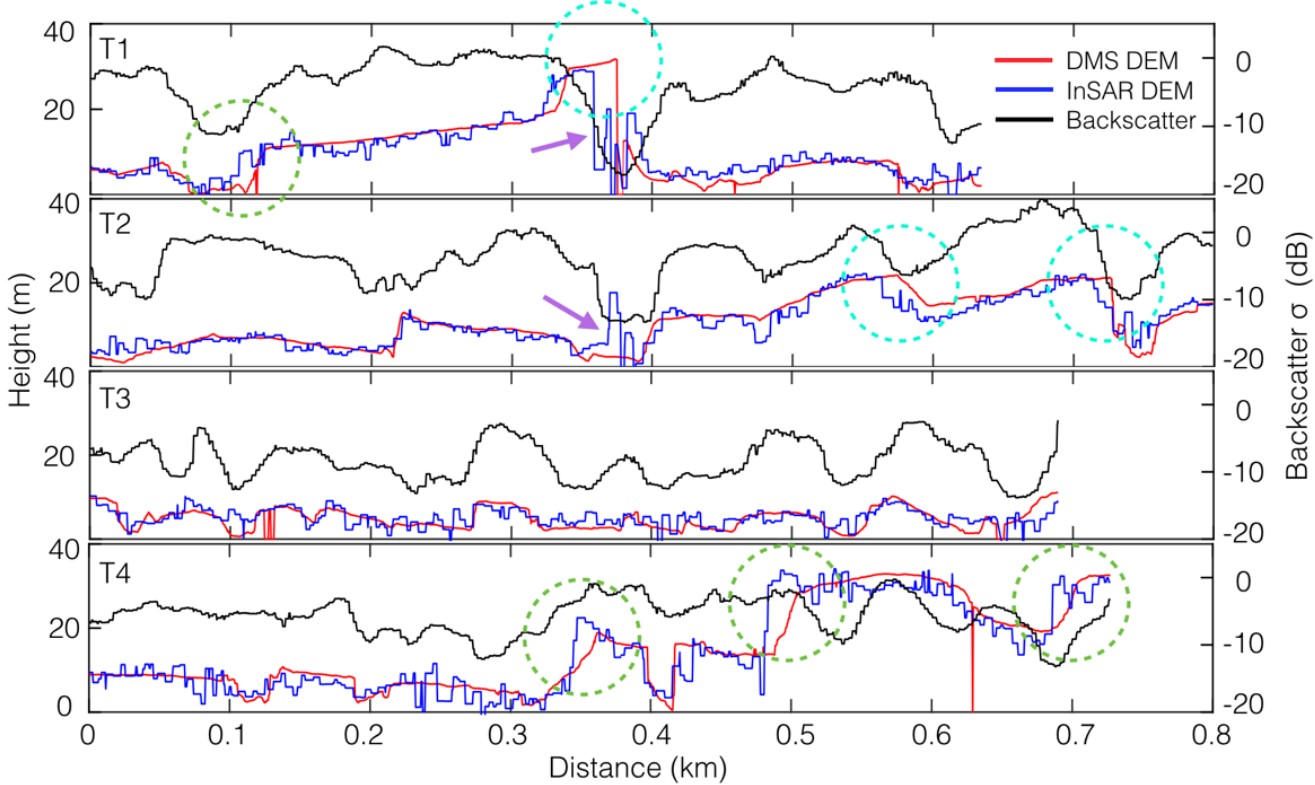

**Figure 5: Comparison of interferometric height (InSAR DEM) (blue), backscatter (black), and OIB DEM (DMS DEM) (red) along transects. Dashed circles indicate locations of significant discrepancy between DEMs either where DMS DEM > InSAR DEM (turquoise) or DMS DEM < InSAR DEM (green). Purple arrows indicate sections of significant (> 10 m) variability in the InSAR DEM not present in the DMS DEM.**





**Figure 6: Correlation between InSAR DEM and DMS DEM for areas A1-A6.**



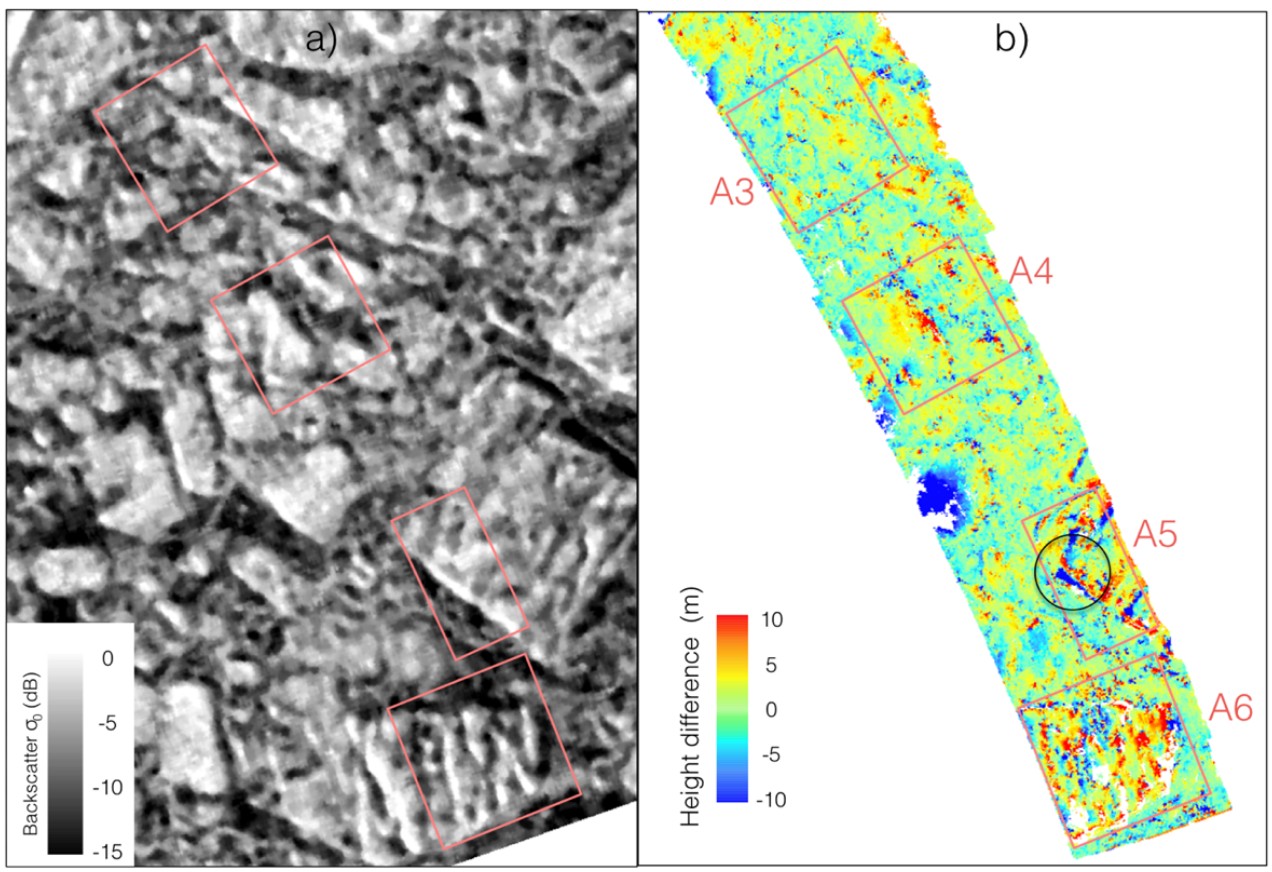

**Figure 7: (a) Intensity cropped around four validation areas. (b) Height difference (InSAR DEM - DMS DEM). Black circle indicates area of substantial difference between DEMs.**





**Figure 8: Iceberg classification based on length scale (a), maximum height (b), volume (c), and tabular height-to-length threshold (d). Buffer Island is masked out in light gray.**





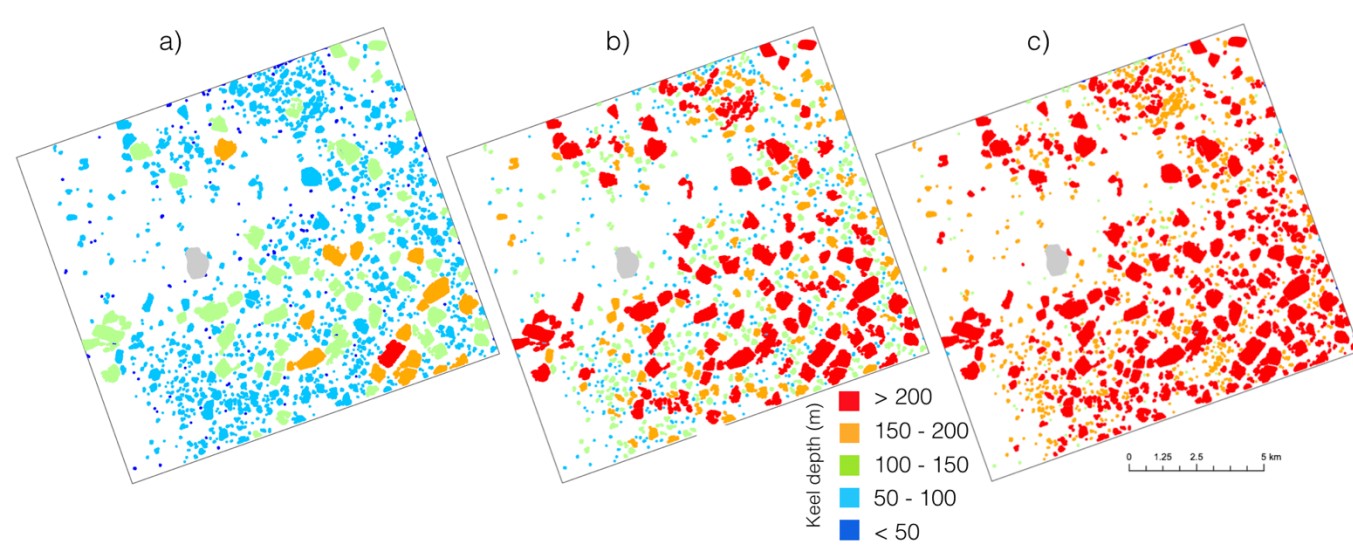

**Figure 9: Derived minimum (a), expected (b), and maximum (c) keel depth.**



Table 1: Iceberg size classification defined by the International Ice Patrol

| Size class | Height (m) | Length (m) |
|:----------:|:----------:|:----------:|
| Growler | < 1 | < 5 |
| Bergy Bit | 1 – 5 | 5 – 15 |
| Small | 5 – 15 | 15 – 60 |
| Medium | 15 – 45 | 60 – 122 |
| Large | 45 – 75 | 122 – 213 |
| Very Large | > 75 | > 213 |




5        Table 2: Volume comparison between InSAR DEM and DMS DEM

| Area | St. dev. height (m) in DMS DEM | Mean height (m) in DMS DEM | InSAR DEM volume ($10^6$ m³) | DMS DEM volume ($10^6$ m³) | Volume difference (%) |
|------|------|------|------|------|------|
| A1 | 1.75 | 3.41 | 0.45 | 0.37 | 22.9 |
| A2 | 7.16 | 10.86 | 0.90 | 0.95 | 5.1 |
| A3 | 2.86 | 5.48 | 0.61 | 0.59 | 3.0 |
| A4 | 4.02 | 7.27 | 0.71 | 0.75 | 5.3 |
| A5 | 10.35 | 13.52 | 1.02 | 1.00 | 2.8 |
| A6 | 8.82 | 14.72 | 1.81 | 1.70 | 6.9 |