# Peer review of "Iceberg topography and volume classification using TanDEM-X interferometry"

_The Cryosphere, 2019_

## Referee Comment (RC1) · Anonymous Referee #1 · 22 Apr 2019

The manuscript investigates the capability of TanDEM-X InSAR to derive iceberg sub-aerial morphology and infer total volume. The validation of InSAR DEM is performed by comparison with airborne (Operation IceBridge) lidar DEM. In Referee's opinion the manuscript can be interesting to the readers but it requires improvements before publishing.

Specific comments and questions: 1. Page 3, line 15: "This work utilizes data from the twin constellation TanDEM-X..." It can be recommended to avoid using "constellation" because it is commonly called as TanDEM-X mission.

2. Referee might suggest to the authors the inclusion of additional recent reference on the topic of the manuscript: I. Zakharov, T. Puestow, D. Power and M. Howell, 2109. "Icebergs in Sea Ice With TanDEM-X Interferometry," in IEEE Geoscience and Remote

[Figure]

Sensing Letters. doi: 10.1109/LGRS.2019.2892896.

3. Page 5, line 15: "The total iceberg volume then can be inferred from this subaerial volume, which is 11% of the total volume..." This number has to be validated. The recent results on iceberg profiling using ship based lidar and sonar demonstrated that sail-to-keel ratio has different values (http://oceansltd.com/iceberg-measurements/ and https://www.onepetro.org/conference-paper/OTC-27473-MS )

4. "In a final alternative approach, we classified icebergs according to minimum, expected, and maximum keel depth (Figure 9)." The results of keel estimation were also not validated in the manuscript. In Referee's opinion the results on keel depth and iceberg total volume have to be validated before publishing.

5. A minimum keel depth as that for tabular icebergs is defined assuming a rectangular cuboid. However, according to Figures 3 and 4 the surface of icebergs has more complicated shape. In this case the question is how accurate the assumption?

6. Page 9, line 30: "Post geocoding, we slightly shifted the DMS DEM (translation) to visually match the InSAR DEM and calibrated the height of both datasets to zero height in an area of no icebergs..." What is the value of shift and inaccuracies such as spatial translational and rotational offsets?

Strength of the manuscript lies in the comparison InSAR DEM with airborne lidar data. However, the authors included results on estimation of iceberg total volume and keel (underwater portion). This is not feasible using InSAR DEM and was not validated in the manuscript. Therefore, I suggest make changes before publishing it.

---

## Referee Comment (RC2) · Anonymous Referee #2 · 23 Apr 2019

The authors appear to have done an insufficient job in performing background research on the topic. In particular, the authors attempt to perform a comprehensive review of the topic of remote sensing and icebergs. The review comes off as being superficial because the authors have missed some very key articles on the subject. For example, in this statement, "Remote sensing techniques such as LiDAR (Scambos et al., 2005) and optical stereo photogrammetry (Enderlin and Hamilton, 2014) have been used to evaluate both the vertical and horizontal extent of icebergs as tools with larger spatial coverage" the authors have missed numerous other citations on the topic. The authors have failed to point out numerous publications that deal with SAR-based iceberg classification. Furthermore, the authors state "However, InSAR has only briefly been explored to acquire information on iceberg topography (Power et al.,2011;Zakharov et

al., 2013;Zakharov et al., 2017)" but they fail to point out that Zakharov et al., 2017 deals specifically with the same topic of bistatic measurement of iceberg topography. More egregious is the omission of the following IEEE publication that seems to be exactly in line with the authors' manuscript.

Zakharov I., Puestow T., Power D. and Howell M. 2019. Icebergs in Sea Ice With TanDEM-X Interferometry, in IEEE Geoscience and Remote Sensing Letters. doi: 10.1109/LGRS.2019.2892896

It is therefore incumbent on the authors to distinguish their manuscript from this already published work. At best, the work is derivative and the manuscript only serves to provide another validation point for this identical technique. Since the manuscript doesn't deal specifically with original research, I believe that the manuscript is perhaps worthy of a conference publication rather than a journal publication.

---

## Referee Comment (RC3) · Anonymous Referee #3 · 7 May 2019

Summary

The paper investigates the capability of single-pass SAR interferometry from TanDEM-X bistatic acquisition for deriving sub-areal morphology and volume of icebergs. The results are validated using Operational IceBridge airborne data. The manuscript is well organized and clear in its explanations. The article yet again demonstrates the capability of the InSAR technique for cryosphere applications. TanDEM-X interferometry has been demonstrated for iceberg (eg. Zakharov et al., 2017)) and sea ice topographic mapping (e.g., Yitayew et al., 2018), and the current article, in particular the capability of deriving volumetric information signifies the potential of the technique for cryosphere applications and hopefully lead to more similar studies over other regions.

I would suggest to consider the following comments before publishing the paper.

[Figure]

General comments: P3, L21. It is noted that the icebergs in the area are "frozen into the landfast sea ice as seen from, . . .". Land fast ice can have topographic features such as ridges which can be as tall as a few meters and that of course influence the classification result presented. Have the authors checked for such structures?

P6, L9. Please comment the significance of the eight-day difference between the acquisitions of the satellite and the validation data on the validation process.

P9, L29-30. ". . . we slightly shifted the DMS DEM (translation) to visually match the InSAR DEM. . .". How accurate is to apply a manual shift (visually matching)? Why can't the geolocation information of both acquisitions be used to accurately align the two measurements? Is this related to the accuracy of the navigational system of the aircraft? Please discuss.

Specific comments:

P2, L23: "... strictly pertaining to the nadir-view two-dimensional shape . . ." Of course, a SAR image is the projection of the 3D info on a 2D plane. However, the scene as imaged by SAR is viewed from an angle (SAR is side looking). Please make it less ambiguous.

P3, L17. I don't think lambda is defined anywhere before that as the wave length. Also, please use "meter-scale" instead of "m-scale" throughout the paper.

P3, L28, [-pi, pi]. (Typo. inverted bracket)

P5, L21. Define L in "d=2.91L^0.71"

P11, L15-16. ". . . High Resolution Side Swatch Mode". Probably should be replaced by "High Resolution Wide Swatch Mode"

---

## Author Comment (AC1) · 12 May 2019

Dear Reviewer 1,

Thank you for reviewing our manuscript and pointing out several items that either were not clear or were missed on our part. We have addressed your specific comments below.

Best regards,
Dyre Dammann

**Review 1:**

The manuscript investigates the capability of TanDEM-X InSAR to derive iceberg subaerial morphology and infer total volume. The validation of InSAR DEM is performed by comparison with airborne (Operation IceBridge) lidar DEM. In Referee's opinion the manuscript can be interesting to the readers but it requires improvements before publishing.

Specific comments and questions:

1. Page 3, line 15: "This work utilizes data from the twin constellation TanDEM-X. . ." It can be recommended to avoid using "constellation" because it is commonly called as TanDEM-X mission.

Done

2. Referee might suggest to the authors the inclusion of additional recent reference on the topic of the manuscript: I. Zakharov, T. Puestow, D. Power and M. Howell, 2109. "Icebergs in Sea Ice With TanDEM-X Interferometry," in IEEE Geoscience and Remote C1 TCD Interactive comment Printer-friendly version Discussion paper Sensing Letters. doi: 10.1109/LGRS.2019.2892896.

We agree. Included now (P3,L10): "TanDEM-X SAR interferometry (InSAR) (Bamler and Hartl, 1998;Rosen et al., 2000) is a technique to extract topography from the phase information from two complex SAR scenes. This technique has previously been utilized to assess sea ice ridges (Dammann et al., 2017;Dierking et al., 2017;Yitayew et al., 2018). For icebergs, this technique was first demonstrated by (García et al., 2012) and later validated by using optical space borne photogrammetry data (Zakharov et al., 2017;Zakharov et al., 2019)."

3. Page 5, line 15: "The total iceberg volume then can be inferred from this subaerial volume, which is 11% of the total volume. . ." This number has to be validated. The recent results on iceberg profiling using ship based lidar and sonar demonstrated that sail-to-keel ratio has different values (http://oceansltd.com/iceberg-measurements/ and https://www.onepetro.org/conference-paper/OTC-27473-MS )

For floating ice, the total volume can be calculated directly from the volume above the waterline (subaerial volume) because the ratio of subaerial volume to total volume only depends on water and ice density. This physics-based approach is also used by Sulak et al., (2017).

As you point out, the sail-to-keel ratio can vary substantially depending on numerous realizations of iceberg size, shape, and 3D spatial and physical characteristics. This is why we provide the maximum and minimum possible keel depths to demonstrate the range of lower to upper limits of keel depths (see next point).

4. "In a final alternative approach, we classified icebergs according to minimum, expected, and maximum keel depth (Figure 9)." The results of keel estimation were also not validated in the manuscript. In Referee's opinion the results on keel depth and iceberg total volume have to be validated before publishing.

We agree this needs to be clarified. As you point out, the real keel depths cannot be validated with the available data (considering the vast statistical ensemble of numerous iceberg shape realizations) Therefore, our maximum and minimum keel depths are based on physics, not observations. We simply derive the maximum and minimum keel depths based on stability of floating objects which are bound by geometrical shapes in the form of a cone and cuboid. We have now restated this as (P5,L26):

"And finally, based on physics of floating objects, we calculate the limits of minimum and maximum keel depths (draft) using idealized shapes and stability analyses (e.g., Bass, 1980)."

We also state more clearly in the discussion (P9,L12):

"In a final alternative approach, we classified icebergs according to physics-based minimum and maximum keel depths as well an estimate based on past data (Figure 9)."

5. A minimum keel depth as that for tabular icebergs is defined assuming a rectangular cuboid. However, according to Figures 3 and 4 the surface of icebergs has more complicated shape. In this case the question is how accurate the assumption?

We agree again that a clarification is needed. We clarify that we derive a window of possible keel depths with lower and upper limits based on physics calculations. This is further clarified (P5,L28):

"We define a physics-based minimum keel depth as that for tabular icebergs $d=V/A$, assuming a rectangular cuboid. We specifically define extreme maximum keel depth (again based on physics of idealized shapes) as an inverted pyramid or cone $d=3V/A$; however, this shape is unlikely to persist due to rapid melting of a pointed keel. Given this definition of maximum keel depth as the upper limit, we note that the window of real-world keel depths is certainly smaller. We also, therefore, estimate "expected" keel depth of $d=2.91L0.71$, where d is the keel depth and L is the waterline length of the iceberg in meters. This approach is suggested by Barker et al. (2004) based on physics combined with limited measurements. A similar analyses was performed by Sulak et al. (2017).

6. Page 9, line 30: "Post geocoding, we slightly shifted the DMS DEM (translation) to visually match the InSAR DEM and calibrated the height of both datasets to zero height in an area of no

icebergs. . .” What is the value of shift and inaccuracies such as spatial translational and
rotational offsets?

We have now specified this better by stating (P10,L14): “A difference in geoid correction
between the datasets can lead to a translational lateral mismatch of up to 80 m, hence we shifted
the DMS DEM 67 m to visually match the InSAR DEM. Based on the strong backscatter
gradients of the iceberg edges a meter-scale accuracy could be ensured.”

Such mismatch is strictly translational and leads to no rotation. However, we provide suggestions
for improvement of this method by using tie-point grids.

Strength of the manuscript lies in the comparison InSAR DEM with airborne lidar data.

We agree, and in addition, we have collocated DEM derived from high-resolution DMS imagery.

However, the authors included results on estimation of iceberg total volume and keel
(underwater portion). This is not feasible using InSAR DEM and was not validated in the
manuscript. Therefore, I suggest make changes before publishing it.

In response to previous points, we have now shown that our analysis is physics-based, rather
than an empirical study. We do agree that with observations we could reduce the range of
possible keel depths. Because we lack these observations, we have chosen to use the maximum
possible range of keel depths based on physics of floating objects.

Dear Reviewer 2,

Thank you for your time and thoughts in reviewing our manuscript. We agree with your suggestions of including more references and placing our work in the context of the new Zakharov letter. We have addressed each of your detailed comments below.

Best regards,
Dyre Dammann

**Review 2:**

The authors appear to have done an insufficient job in performing background research on the topic. In particular, the authors attempt to perform a comprehensive review of the topic of remote sensing and icebergs. The review comes off as being superficial because the authors have missed some very key articles on the subject. For example, in this statement, "Remote sensing techniques such as LiDAR (Scambos et al., 2005) and optical stereo photogrammetry (Enderlin and Hamilton, 2014) have been used to evaluate both the vertical and horizontal extent of icebergs as tools with larger spatial coverage" the authors have missed numerous other citations on the topic.

The authors have failed to point out numerous publications that deal with SAR-based iceberg classification.

We have now included more references providing better overview.

Furthermore, the authors state "However, InSAR has only briefly been explored to acquire information on iceberg topography (Power et al.,2011;Zakharov et al 2013;Zakharov et al., 2017)" but they fail to point out that Zakharov et al., 2017 deals specifically with the same topic of bistatic measurement of iceberg topography. More egregious is the omission of the following IEEE publication that seems to be exactly in line with the authors' manuscript.

Zakharov I., Puestow T., Power D. and Howell M. 2019. Icebergs in Sea Ice With TanDEM-X Interferometry, in IEEE Geoscience and Remote Sensing Letters. doi: 10.1109/LGRS.2019.2892896

We initially missed the 2019 letter as this was coming out when our paper was completed and only a few weeks before our work was submitted. We have now clearly placed this work in context of work by C-CORE (P3,L10):

"TanDEM-X SAR interferometry (InSAR) (Bamler and Hartl, 1998;Rosen et al., 2000) is a technique to extract topography from the phase information from two complex SAR scenes. This technique has previously been utilized to assess sea ice ridges (Dammann et al., 2017;Dierking et al., 2017;Yitayew et al., 2018). For icebergs, this technique was first demonstrated by (García et

al., 2012) and later validated by using optical space borne photogrammetry data (Zakharov et al., 2017;Zakharov et al., 2019)."

and highlighting the scope of this paper and how it differs from prior work by stating (P3,L15):

"The work presented here expand upon prior work. First, we validate this technique using high-resolution airborne data from optical imagery in combination and cross-validated with laser altimeters resulting in ~20 cm vertical accuracy. These missions were carefully planned with the German Aerospace Center for collocating TanDEM-X data acquisitions with both optical imagery and laser altimetry forming comprehensive coordinated datasets. Second, we demonstrate possible applications and uses of this approach. We present and validate an alternate method to standard iceberg classification based on volume. Volume is arguably more relevant than standard measurements such as height and length for applications including freshwater contribution, drift, and potential impact on structures. We also investigate the derivation of possible iceberg keel depths relevant for grounding assessments and impact on subsea installations. Lastly, we include a discussion around necessary considerations including phase noise, signal penetration, and acquisition geometry. We also highlight potential limitations related to data availability, iceberg shape and size, and the impacts of drift."

It is therefore incumbent on the authors to distinguish their manuscript from this already published work. At best, the work is derivative and the manuscript only serves to provide another validation point for this identical technique. Since the manuscript doesn't deal specifically with original research, I believe that the manuscript is perhaps worthy of a conference publication rather than a journal publication.

We have now placed this work in view of the Zakharov et al. (2019) published as a 5-page letter. We agree that this letter does provide a valuable insight into bistatic InSAR as an approach to assess iceberg morphology. However, we disagree that the letter should prevent this manuscript to be published. The Zakharov et al. (2019) letter omits significant details and a comprehensive discussion around applications and limitations. In our work, we apply a different validation technique drawing upon high-resolution combined optical and laser scanning allowing for cross-validation in coordination with collocated TanDEM-X data. These datasets were not available in prior studies. The U.S. National Ice Center (forming the North American Ice Service together with the Canadian Ice Service and the International Ice Patrol responsible for iceberg detection and monitoring) that had recognized the potential value of OTASC for operational sea ice and iceberg applications, and thereby participated, supported, and contributed to the success of OTASC (Nghiem et al., 2018). We also take this work further by demonstrating the application of classifying icebergs based on volume and keel depths with direct implications for ice users. Lastly, we are providing an in-depth, broad and much needed discussion around this topic. This is now better clarified in the introduction (see above).

Dear Reviewer 3,

Thank you for reviewing our manuscript. Your suggestions have helped improve this paper. Please see responses to each of your comments below.

Best regards,
Dyre Dammann

The paper investigates the capability of single-pass SAR interferometry from TanDEMX bistatic acquisition for deriving sub-areal morphology and volume of icebergs. The results are validated using Operational IceBridge airborne data. The manuscript is well organized and clear in its explanations. The article yet again demonstrates the capability of the InSAR technique for cryosphere applications. TanDEM-X interferometry has been demonstrated for iceberg (eg. Zakharov et al., 2017)) and sea ice topographic mapping (e.g., Yitayew et al., 2018), and the current article, in particular the capability of deriving volumetric information signifies the potential of the technique for cryosphere applications and hopefully lead to more similar studies over other regions.

I would suggest to consider the following comments before publishing the paper.

General comments:

P3, L21. It is noted that the icebergs in the area are "frozen into the landfast sea ice as seen from, . . .". Land fast ice can have topographic features such as ridges which can be as tall as a few meters and that of course influence the classification result presented. Have the authors checked for such structures?

This is a good point. Sea ice can indeed feature ridges on the scales of several meters. In this work we only classify ice exceeding 5 m. Ridges exceeding 5 m is rarer, but can certainly happen. However, such elevations will likely not persist on large spatial scales. We perform a geometric opening that removes much of the small clusters of pixels where elevation exceed 5 m, which is expected to remove the signal from sea ice ridges. With that said also, we have not noticed an obvious presence of large landfast sea ice ridges in our data likely partly due to the sheltered location of the bay.

P6, L9. Please comment the significance of the eight-day difference between the acquisitions of the satellite and the validation data on the validation process.

Good point. We have now moved the comment about this down from an earlier subsection. We agree this needs to be comment on here, since here is where it is relevant. We now state (P6,L26):

"No significant motion took place between the two datasets. The icebergs were confirmed stationary, frozen into the landfast ice, by consecutive TanDEM-X overpasses."

P9, L29-30. ". . . we slightly shifted the DMS DEM (translation) to visually match the InSAR DEM. . .". How accurate is to apply a manual shift (visually matching)? Why can't the geolocation information of both acquisitions be used to accurately align the two measurements? Is this related to the accuracy of the navigational system of the aircraft? Please discuss.

Thank you for pointing this out. We acknowledge that this is not the optimal way of matching the datasets. We now discuss that this is due to a mismatch between the geoid correction of the two datasets (P10,L14):

"A difference in geoid correction between the datasets can lead to a translational lateral mismatch of up to 80 m, hence we shifted the DMS DEM 67 m to visually match the InSAR DEM.

We also comment that this only leads to meter-scale accuracy (P10,L15):

"Based on the strong backscatter gradients of the iceberg edges a meter-scale accuracy could be ensured."

These included sentences now compliment the rest of the paragraph highlighting ways to improve this analysis by using tie-point grids to ensure higher accuracy for future work.

Specific comments:

P2, L23: "... strictly pertaining to the nadir-view two-dimensional shape . . ." Of course, a SAR image is the projection of the 3D info on a 2D plane. However, the scene as imaged by SAR is viewed from an angle (SAR is side looking). Please make it less ambiguous.

Simplified this to (P2,L29): "However, standard SAR-based approaches provide information strictly pertaining to the horizontal extent and concentration of icebergs, but do not provide information pertaining to the height necessary to fully classify the icebergs according to Table 1."

P3, L17. I don't think lambda is defined anywhere before that as the wave length. Also, please use "meter-scale" instead of "m-scale" throughout the paper.

We have now moved this specification down to after lambda has been defined. m-scale has been changed to meter-scale.

P3, L28, [-pi, pi]. (Typo. inverted bracket)

Changed

P5, L21. Define L in "d=2.91L^0.71"

This has been explained now as the waterline length

P11, L15-16. ". . . High Resolution Side Swatch Mode". Probably should be replaced by "High Resolution Wide Swath Mode"

Yes, thank you for catching this. Changed to "High Resolution Wide Swath Mode"

[revised manuscript text omitted]

---

## Author Comment (AC3) · 6 Jun 2019

The comment was uploaded in the form of a supplement:
https://www.the-cryosphere-discuss.net/tc-2019-59/tc-2019-59-AC3-supplement.pdf

---

## Author Comment (AC4) · 6 Jun 2019

The comment was uploaded in the form of a supplement:
https://www.the-cryosphere-discuss.net/tc-2019-59/tc-2019-59-AC4-supplement.pdf